# Development and Optimization of an Enzyme Immunoassay to Detect Serum Antibodies against the Hepatitis E Virus in Pigs, Using Plant-Derived ORF2 Recombinant Protein

**DOI:** 10.3390/vaccines9090991

**Published:** 2021-09-06

**Authors:** Katerina Takova, Tsvetoslav Koynarski, George Minkov, Valentina Toneva, Eugenia Mardanova, Nikolai Ravin, Georgi L. Lukov, Gergana Zahmanova

**Affiliations:** 1Department of Plant Physiology and Molecular Biology, University of Plovdiv, Plovdiv 4000, Bulgaria; katerina.takova@uni-plovdiv.bg (K.T.); george-minkov@uni-plovdiv.bg (G.M.); toneva@plantgene.eu (V.T.); 2Department of Animal Genetics, Faculty of Veterinary Medicine, Trakia University, Stara Zagora 6000, Bulgaria; tkoynarski@gmail.com; 3Institute of Molecular Biology and Biotechnologies, Markovo 4108, Bulgaria; 4Institute of Bioengineering, Research Center of Biotechnology of the Russian Academy of Sciences, Moscow 119071, Russia; nravin@mail.ru (N.R.); mardanovaes@mail.ru (E.M.); 5Faculty of Sciences, Brigham Young University—Hawaii, Laie, HI 96762, USA; georgi.lukov@byuh.edu; 6Center of Plant Systems Biology and Biotechnology, Plovdiv 4000, Bulgaria

**Keywords:** hepatitis E virus, detection of serum antibodies, transient expression, plant molecular farming, ORF2 capsid protein, diagnostic antigen

## Abstract

Hepatitis E is an emerging global disease, mainly transmitted via the fecal–oral route in developing countries, and in a zoonotic manner in the developed world. Pigs and wild boar constitute the primary Hepatitis E virus (HEV) zoonotic reservoir. Consumption of undercooked animal meat or direct contact with infected animals is the most common source of HEV infection in European countries. The purpose of this study is to develop an enzyme immunoassay (EIA) for the detection of anti-hepatitis E virus IgG in pig serum, using plant-produced recombinant HEV-3 ORF2 as an antigenic coating protein, and also to evaluate the sensitivity and specificity of this assay. A recombinant HEV-3 ORF2 110-610_6his capsid protein, transiently expressed by pEff vector in *Nicotiana benthamiana* plants was used to develop an in-house HEV EIA. The plant-derived HEV-3 ORF2 110-610_6his protein proved to be antigenically similar to the HEV ORF2 capsid protein and it can self-assemble into heterogeneous particulate structures. The optimal conditions for the in-house EIA (iEIA) were determined as follows: HEV-3 ORF2 110-610_6his antigen concentration (4 µg/mL), serum dilution (1:50), 3% BSA as a blocking agent, and secondary antibody dilution (1:20 000). The iEIA developed for this study showed a sensitivity of 97.1% (95% Cl: 89.9–99.65) and a specificity of 98.6% (95% Cl: 92.5–99.96) with a Youden index of 0.9571. A comparison between our iEIA and a commercial assay (PrioCHECK™ Porcine HEV Ab ELISA Kit, ThermoFisher Scientific, MA, USA) showed 97.8% agreement with a *kappa* index of 0.9399. The plant-based HEV-3 ORF2 iEIA assay was able to detect anti-HEV IgG in pig serum with a very good agreement compared to the commercially available kit.

## 1. Introduction

Hepatitis E viral (HEV) infections are currently the most common cause of acute hepatitis in the world. The vast majority of Hepatitis E cases present as a self-limited acute illness, which in rare cases, primarily in immunocompromised individuals, can become chronic. [1]. The overall case-fatality rate is about 1%, but in pregnant women in their third trimester, it can reach up to 30% [2].

HEV is a quasi-enveloped (in blood) and non-enveloped (in feces), positive-sense single-stranded RNA virus [3], classified in the *Hepeviridae* family, genus *Orthohepevirus*. The genus *Orthohepevirus* contains four species (A through D) [4]. *Orthohepevirus A* has been considered to infect humans, pigs, wild boars, deer, camels, mongooses, and rabbits. *Orthohepevirus B* infects avians, *Orthohepevirus C* contains viruses infecting rats and ferrets, and *Orthohepevirus D* infects bats [5]. There are eight genotypes within the *Orthohepevirus A* species (HEV-1 to HEV-8), with the first four commonly infecting humans. HEV-1 and HEV-2 only infect humans and are transmitted via the fecal–oral route, while HEV-3 and HEV-4 are zoonotic and can infect both humans and animals [6]. HEV-3 is the most prevalent genotype in Europe, with three major clades: HEV-3abjchi, HEV-3efg (found in humans, pigs, wild boars, and deer), and HEV-3ra (found in rabbits) [7,8,9,10]. The HEV genome contains three open readings frames (ORF), which encode structural and non-structural proteins [11,12]. The ORF2 capsid protein is a major antigenic and immunogenic protein, containing neutralizing antibodies inducing epitopes [13]. It is also an appropriate diagnostic antigen of HEV [14,15,16,17].

HEV infection is fecal-orally transmitted by contaminated water in developing countries [18]. Zoonotic and blood transfusions are more common routes in industrialized countries [19,20,21,22]. HEV spreads from animals to humans through the consumption of undercooked pork meat and exposure when handling animals, particularly pigs. The reported HEV seroprevalence in swine herds in Europe ranges between 30% and 100%, with similar results observed in Bulgarian herds as well [15,23,24,25,26,27]. Overall HEV seroprevalence in wild boar is reported to be between 12% and 46% [25,28,29]. These investigations have shown that domestic pigs and wild boar are the main zoonotic reservoirs of HEV. It is therefore necessary to regularly monitor the spread of HEV infection in zoonotic vectors, as well as to develop an accessible, reliable, and accurate immunoassay for HEV detection.

HEV infection diagnosis in pigs is based on the detection of specific IgM and IgG anti-HEV antibodies, as well as of HEV RNA. Serological HEV diagnostic tests are commercially available (PrioCHECK™ Porcine HEV Ab ELISA Kit, ThermoFisher Scientific, Waltham, MA, USA; ID Screen^®^ Hepatitis E Indirect Multi-species, Grabels, France). These tests use recombinant genotype 3 capsid proteins produced by expensive methods, which makes mass screening of HEV distribution among animals cost-prohibitive. Thus, higher-yield and more cost-effective methods for recombinant HEV protein production are urgently needed, for use as diagnostic antigens in the serological diagnosis of HEV.

Different regions of the ORF2 capsid protein have been expressed in different expression systems, including *E. coli* [30,31], yeasts [32,33], insect cells [34], mammalian cells [35], and plants [36,37]. Although good expression has been achieved in all expression systems, plants offer many advantages such as a low risk of contamination by human/animal pathogens, the ability to perform eukaryotic post-translational modification, and easy downstream purification, as well as process scalability [38,39]. The accumulation of large amounts of recombinant proteins in plant expression systems has been achieved through the development of different transient expression systems based on plant viral vectors [40,41,42]. An example of a viral-based vector is the pEAQ-HT vector, which is based on the Cowpea mosaic virus (CPMV). It is a non-replicating system that uses 5′-UTR and 3′-UTR from CPMV RNA-2 to achieve high level expression [43]. Another example is the potato X virus (PVX)-based self-replicating vector pEff. This is comprised of the 5′-UTR of the PXV genome, the gene for the RNA-dependent RNA polymerase, the first promoter of subgenomic RNAs, the AMV translation enhancer (5′-UTR of the alfalfa mosaic virus RNA), and the 3′-UTR of the PVX genome [41]. Both systems have been used for the expression of a HEV-3 ORF2 110-610_6his capsid protein, and the pEff vector showed a higher level of accumulation of the HEV capsid protein compared with the pEAQ-HT vector [37]. 

Plant-derived antigens have demonstrable utility in the serological detection of a number of infectious diseases, such as the Dengue virus [44,45], West Nile Virus [46], Rift Valley fever virus [47], and Hepatitis E virus [15]. These studies demonstrate that plants are among the most rapid, prolific, and flexible producers of diagnostic immunogenic reagents. 

Recently, the recombinant HEV-3 ORF2 110-610_6his was transiently expressed and characterized in plants, showing its immunogenic [48] and diagnostic potential [15]. In this study, we developed and optimized an in-house EIA designed to detect HEV serum antibodies in pigs, using the plant-derived HEV-3 ORF2 110-610_6his recombinant protein. This plant-produced protein proved to be antigenically similar to the HEV capsid protein and suitable for pig serum screening for anti-HEV antibody prevalence evaluation.

## 2. Materials and Methods

### 2.1. Gene Cloning, Plasmids Construction, and Agroinfiltration of Nicotiana Benthamiana

The target HEV-3 ORF2 110–610 gene (GenBank accession number DQ079627.1) was produced as previously described by our group [37]. Briefly, the plasmid pEff:HEV-3 ORF2 110-610_6his was transformed into competent *Escherichia coli* XL1Blue; putative clones harboring the expression vector (pEff:HEV-3 ORF2 110-610_6his) were verified by sequencing (Eurofins, Hamburg, Germany). Recombinant vectors were transformed into the electrocompetent *Agrobacterium tumefaciens* strain GV3101. These bacteria were cultured in a Luria–Bertani medium containing 50 μg/mL Kanamycin and 100 μg/mL Rifampicin, for 16 hours at 28 °C. The culture was then pelleted. The obtained bacteria were resuspended in an infiltration solution containing 10 mM MES 2-(N-morpholino) ethansulfonic acid with a pH of 5.5, 10 mM MgSO_4_, and 100 µM acetosyringone.

Leaves of 5–6 week-old *N. benthamiana* plants were infiltrated with a suspension (OD600 ~0.2) of recombinant pEff:HEV-3 ORF2 110-610_6his *A. tumefaciens* strain GV3101 using a syringe. Leaves were harvested at 4 days post infiltration (dpi).

### 2.2. SDS-PAGE and Western Blot of HEV-3 OFR2 110-610_6his Protein Preparations

Small-scale protein extraction was conducted to test the expression levels of the pEff:HEV-3 ORF2 110-610_6his vector. Leaf discs from the infiltrated zones (~90 mg) were excised and homogenized in 3× volume of extraction buffer (0.4 M sucrose, 50 mM Tris (pH 8.0), 5 mM MgCl_2_, 10% glycerol, 5 mM β-mercaptoethanol). The resulting mixture was centrifuged at 13,000g for 10 min, and the supernatant was taken. An equal volume of 2× sample buffer was added to the supernatant. Then, 10 μL of the resulting mixture (corresponding to 1.5 mg of fresh leaf tissue) was analyzed by SDS-PAGE. After electrophoresis, the gel was stained with One-Step Blue® Protein Gel Stain (BIOTIUM, Fremont, CA, USA). The intensity of the bands in stained gels was determined using the Nonlinear Dynamics TotalLab TL120v2009-NULL software.

For the Western blot assay, the proteins were transferred from the SDS-PAGE gel onto a nitrocellulose membrane (Bio-Rad Laboratories Ltd, Hertfordshire, UK). Membranes were blocked with 5% (w/v) non-fat dried milk in PBS with 0.05% Tween-20 (v/v) (PBST) and incubated for 1 h with a mouse anti-hepatitis E ORF2 antigen primary antibody (ab101124; Abcam, Cambridge, UK), diluted 1:1 000 at room temperature and washed with PBST. The bound antibody was detected with secondary anti-mouse antibody-horseradish peroxidase (HRP) (ThermoFisher Scientific, Waltham, MA, USA), diluted 1:30,000. The emitted luminescence from the ECL detection reagents (GE Healthcare Life Sciences, Buckinghamshire, UK) was detected with the ChemiDoc Imager (Bio-Rad, Hercules, CA, USA). For Western blot analysis using pig serum as the primary antibodies, the membrane was incubated with pig serum (dilution 1:50), and bound antibodies were detected with secondary anti-swine IgG Phosphatase-Labeled (KPL, Gaithersburg, MD, USA), diluted 1:10,000, respectively. One-step NBT/BCIP substrate (ThermoFisher Scientific, Waltham, MA, USA) was used.

### 2.3. Large-Scale Isolation of HEV-3 ORF2 110-610_6his Recombinant Proteins from Plant Tissue

Large-scale sampling was conducted by removing any non-infiltrated tissue and recording leaf sample weight. Samples were extracted by adding 3 volumes of the PBS pH 7.2 extraction buffer with complete EDTA-free protease inhibitor cocktail tablets (Roche Diagnostics GmbH, Mannheim, Germany), and was then mechanically homogenized. The HEV-3 ORF2 110-610_6his protein carrying the C-terminal 6-histidine tag was isolated under native conditions, using immobilized metal-anion chromatography (IMAC) on a Ni-NTA column, according to the manufacturer’s instructions (Qiagen, Hilden, Germany). The eluted protein was dialyzed against PBS pH 7.2, and four buffers were changed using Slide-A-Lyzer Mini dialysis devices (Thermo Fisher Scientific, Waltham, MA, USA). Proteins were quantitated by a Qubit Protein Assay Kit, using Qubit Fluorometer (Thermo Fisher Scientific, Waltham, MA, USA), following the manufacturer’s instructions.

### 2.4. Transmission Electron Microscopy (TEM), Cryo-EM, Atomic Force Microscopy

A total of 3 μL of purified HEV-3 ORF2 110–610_6his recombinant protein was adsorbed onto glow discharged Carbon-Formvar-coated copper grids for 60 s. The grids were then washed by floating on water droplets, followed by staining with 2% (*w*/*v*) uranyl acetate (UA) for 20 s. Particles were imaged using an FEI Tecnai G2 20 (FEI, Hillsboro, OR, USA) electron microscope operating at 200 kV. An Integra Prima microscope and Nova SPM software (NT-MDT, Moscow, Russia) were used for atomic force microscopy. Scanning was performed in semi-contact mode, using gold cantilever NSG01 (NT-MDT). The protein sample was applied to a sapphire substrate coated with mica and dried at room temperature. PBS was used as a negative control. Cryo-EM grids were prepared as described by Byrne et al., 2019 [49].

### 2.5. Serum Samples

A panel of 141 serum samples (69 HEV IgG positive and 72 negative) from pigs (*Sus scrofa domesticus*) was used to determine the cut-off value and performance of the iEIA (specificity, sensitivity, efficiency, and Youden index values). The study was approved by the IMBB institutional ethic committee (process number EK-03022021).

### 2.6. Optimization of iEIA Protocol Based on HEV-3 ORF2 110-610_6his Recombinant Protein

A total of 5 negative and 5 positive serum samples, randomly selected from previously evaluated samples, were tested in three independent experiments. Optimal working dilutions of the coating HEV-3 ORF2 110-610_6his recombinant protein, pig serum, and secondary antibody horseradish peroxidase (HRP) conjugate have been determined by checkerboard titration. Briefly, well plates were coated with 0.5, 1, 2, 4, and 8 µg/mL of the recombinant protein in Bicarbonate/carbonate coating buffer (100 mM) with a pH of 9.6. The microtiter plates (Greiner 96-well flat bottom) were coated with 50 µL/well of purified coating protein and incubated overnight at 4 °C. After three washes with PBST, the plates were incubated with 200 µL/well of blocking solution (PBST–3% [*w*/*v*] BSA) for 1 h at room temperature. Aliquots of pig sera diluted at 1:50, 1:100, 1:200, and 1:400 in blocking buffer were dispensed into the wells of the plates and incubated for one hour at 37 °C. After three washes with PBST, HRP-conjugated goat anti-swine secondary antibody (ThermoFisher Scientific, Waltham, MA, USA) was added at a dilution of 1:5000, 1:10,000, and 1:20,000. After incubation with the secondary antibody, plate wells were washed three times before 50 µL/well of the substrate solution (*o*-phenylenediamine, Millipore–Sigma, Munich, Germany) was added. Plates were incubated in the dark at room temperature for 20 min. The reaction was stopped with 1 M H_2_SO_4_, and the plates were read at 492 nm in an Epoch Microplate Spectrophotometer plate reader (BioTek Instruments Inc., Winooski, VT, USA). Mean positive/negative (P/N) ratio was calculated.

### 2.7. Validation, Statistical Analysis, and Cut-off Evaluation of the Developed iEIA Performance

NCSS 2021 Data Analysis software was used to find the most appropriate cut-off value, receiver operated characteristic (ROC), specificity, sensitivity, efficiency, Youden index values, and area under the curves (AUCs).

Specificity = no. of true negative/(no. of true negative + no. of false positive)Sensitivity = no. of true positive/(no. of true positive + no. of false negative)Efficiency (%) = ((no. of true positive + no. of true negative)/(no. of true positive + no. of false positive + no. of true negative + no. of false negative)) × 100Youden index = sensitivity + specificity − 1

The agreement between the commercial immunoassay test, PrioCHECK™ Porcine HEV Ab ELISA Kit, (ThermoFisher Scientific, USA), and the in-house assays was assessed by pairwise comparisons using the *kappa* coefficient. The Cohen’s *kappa* is a statistical coefficient, which represents the degree of accuracy and reliability in statistical classification. It measures the agreement between two raters classifying items into mutually exclusive categories. *Kappa* = (P_0_ − P_E_)/(1 − P_E_), where Po is the relative observed agreement among raters, and Pe is the hypothetical probability of chance agreement.

## 3. Results

### 3.1. Production and Purification of HEV-3 ORF2 110-610_6his

We used pEff vector in order to achieve HEV-3 ORF2 capsid protein production in *Nicotiana benthamiana* plants. An HEV genotype 3 capsid protein consisting of amino acid residues 110 to 610 was cloned into pEff to make a recombinant viral vector pEff:HEV-3 ORF2 110-610_6his [37]. To facilitate purification of the HEV-3 ORF2 110-610 capsid protein, a 6-histidine tag was added to the C-terminus of the recombinant protein. *N. benthamiana* leaves, agroinfiltrated with the recombinant vector pEff:HEV-3 ORF2 110-610_6his, were harvested on the 4th day post infiltration (dpi) and then analyzed by SDS-PAGE and Western blot (Figure 1, Appendix A). The lysates from leaves infiltrated with the recombinant protein produced a clear band with the expected molecular weight (~56 kDa; Figure 1a), which was not present in leaves inoculated with the empty pEff vector. Western blot analysis with an anti-HEV ORF2 monoclonoal antibody (mAb) (Figure 1b) confirmed that the HEV-3 ORF2 110-610_6his protein had been expressed well in *N. benthamiana* plants. Pig serum, positive for anti-HEV IgG, also successfully recognized the recombinant protein during Western immunoblotting (Figure 1c).

The HEV-3 ORF2 110-610_6his protein was isolated from *N. benthamiana* leaves using metal-affinity chromatography under native conditions. After purification, the protein samples were dialyzed against phosphate-buffered saline (PBS) and analyzed by SDS-PAGE and Western blot with a monoclonal anti-ORF2 antibody (Figure 2 and Appendix A). The IMAC method applied here obtained the ~200 µg HEV-3 ORF2110-610_6his/g of fresh tissue.

### 3.2. In Vitro Particles Formation

Samples of IMAC purified, plant-produced HEV-3 ORF2 110-610_6his protein were subject to atomic force microscopy (Figure 3a) and transmission electron microscopy (Figure 3b,c). Spherical particles with heterogeneous size, with a diameter of approximately 18 ± 8 nm, were observed in the dialyzed fractions (Figure 3). Additionally, the particles were analyzed using cryo-EM (Figure 3c), though we were unable to reconstruct the 3D structure of the particles due to their heterogeneity. The results revealed that purified HEV-3 ORF2 110-610_6his protein can self-assembled into particulate structures in vitro with a very heterogeneous size.

### 3.3. In-House EIA Optimization Using the Plant-Derived HEV-3 ORF2 110-610_6his as a Coating Protein

The iEIA was optimized using different antigen concentrations, serum dilutions, secondary antibody dilutions, and blocking agents. We used 3% BSA in PBST as the blocking agent. To determine the optimal antigen coating concentration, we used 0.5, 1, 2, 4, and 8 µg/mL HEV-3 ORF2 110-610_6his. The used serum dilutions were 1:50, 1:100, 1:200, and 1:400, and the secondary antibody dilutions were 1:5000, 1:10,000, and 1:20,000. We determined the optimal relation between the positive and the negative samples (P/N) in all combinations between the coating antigen, serum dilutions, and second antibody using a 3% BSA blocking agent (Figure 4). The optimal concentration of recombinant capsid protein used to coat the iEIA plate was 4 µg/mL. The optimal serum and secondary antibody dilutions, used with the 4 µg/mL antigen concentration, were 1:50 and 1:20,000, respectively. These conditions produced the highest positive/negative ratio for the standard checkerboard titration. 

### 3.4. Specificity and Sensitivity of the In-House Enzyme Immunosorbent Assay with the Plant-Derived HEV ORF2 110-610_6his Protein

The cut-off value and sensitivity of the iEIA were determined by testing 72 negative and 69 positive pig serum samples for antibodies against HEV, as determined by the PrioCHECK™ Porcine HEV Ab ELISA Kit. The Receiver Operating Characteristic (ROC) curve (Figure 5) had an area the under curve (AUC) value of 0.9942 (95% Cl: 0.9672–0.999), sensitivity of 97.1% (95% Cl: 89.9–99.65), and specificity of 98.6% (95% Cl: 92.5–99.96) (Table 1). Assay cut-off value (A_492_ nm = 0.92) was determined as the optimal values of the sensitivity and specificity.

Comparison between the commercial PrioCHECK™ Porcine HEV Ab ELISA Kit and the in-house EIA revealed strong agreement between the results of the two methods, with 97.8% accuracy and a *kappa* index of 0.93. Of the 141 tested pig serums, the commercial kit identified 69 positive and 72 negative samples. Of the 69 positive samples, 67 were also identified as positive by the iEIA method (sensitivity of 97.1%). The remaining 2 we identified as negative. When the negative samples were tested with the iEIA test, 71 of the 72 samples were also identified as negative (specificity of 98.6%). Only one of the commercial-kit-identified negative samples was identified as positive by the in-house method. Overall, only 3 of the 141 tested samples showed conflicting results, demonstrating a strong correlation between iEIA and the commercial PrioCHECK™ Porcine HEV Ab ELISA Kit (Table 1).

## 4. Discussion

The Hepatitis E virus is an emerging zoonotic disease in most developed countries, fast becoming a major concern for health authorities. Screening for HEV seroprevalence in pigs (the primary animal reservoir) should help in monitoring virus spread, as well as aid in taking adequate measures to limit transfection. Multiple investigations have shown that domestic pigs and wild boar in Bulgaria are widely infected with HEV [23,24,25,26,27]. Serological immunoassay development offers a highly sensitive and cost-effective diagnostic tool for HEV detection in animals [14,50,51]. 

We investigated the potential use of plants in HEV-3 ORF2 capsid protein production, which was then used as an immunoassay antigen for HEV serum antibody detection in pigs. Plants have been used as expression systems for rapid and low-cost production of recombinant immunogenic proteins, utilized to detect and diagnose a number of viral diseases, and for vaccine production. The concept of plant-based vaccines found its commercial application when the United States Department of Agriculture approved a vaccine against the Newcastle disease. This led to an increased production of recombinant proteins in plants as reagents for diagnostic tools or vaccines [52,53,54,55].

The ORF2 capsid protein is the major immunogenic protein inducing neutralizing antibodies, and the main antigen used in serological HEV diagnosis. All commercially available serological diagnostic kits use the ORF2 capsid protein for HEV infection detection, either alone or in combination within the ORF3 protein [56,57]. The different HEV genotypes have common epitopes within the ORF2 capsid protein, making capsid proteins with animal and human origin equal for detection of anti-HEV antibodies by immunoassays [58]. Better immunoassay immunoreactivity could be achieved with coating antigens belonging to the different HEV genotypes [59].

Here, we developed and optimized an HEV iEIA for the detection of serum IgG, based on the recombinant HEV-3 ORF2 110-610_6his capsid protein produced in *N. benthamiana* plants. Our previous studies describe the successful expression of the HEV-3 ORF2 110-610 gene [37,48] using the Cowpea Mosaic Virus (CPMV)-based vector pEAQ-HT [Sainsbury 2008 and 2009] and the potato X virus (PVX)-based vector pEff [41]. The latter showed higher HEV-3 ORF2 110-610_6his protein accumulation levels as compared to the CPMV-based vector pEAQ-HT [37], with a yield up to ~200 µg/g of fresh weight (FW) plants leaves. This makes the pEff vector preferable in a system for HEV-3 ORF2 gene expression in *N. benthamiana.* The plant-derived HEV-3 ORF2 110-610_6his protein was successfully recognized by the polyclonal anti-HEV IgG Ab in pig sera (Figure 1), which shows that plants produced antigenically similar protein to the HEV ORF2 capsid. Moreover, the plant-produced HEV-3 ORF2 110-610_6his protein was easily purified using the IMAC method with high purity and productivity (Figure 2). The purified recombinant HEV-3 ORF2 110-610_6his protein gave a clear band with a molecular mass of around 56 kDa (Figure 2). The sensitivity and specificity of the iEIA are dependent on the high purity of the used coating protein, thus optimal HEV-3 ORF2 protein purity is preferable.

The expression of N-terminal and a C-terminal truncated forms of the ORF2 capsid protein in *N. benthamiana* was previously shown to tolerate its self-assembly into VLPs [37]. In our study, the HEV capsid protein formed particulate structures (Figure 3) with conformational epitopes when the HEV-3 ORF2 110-610_6his antigen was used under native conditions. The major anti-HEV antibody responses are against epitopes located in HEV ORF2, which are mainly conformational [60,61]. The use of HEV-3 VLPs as a coating antigen gives us several advantages, such as coating protein stability and detection of antibodies interacting with the conformational epitopes.

In this study, we found that the optimal concentration of antigen was set at 4 µg/mL, with a final volume used in the assay of 50 μL (equal to 0.2 µg/well). The amount of 20 µg is needed for coating a whole 96-well plate. This means ~900 unique samples can be tested with the recombinant protein gained from 1 g of fresh leaf mass. The optimal antigen concentration in this study is higher than we reported previously [15], when 0.1 µg/well was used. In many studies, however, higher antigen concentrations (0.67 µg/well) have been used in ELISA for HEV [17]. It should be taken into consideration that different studies used microplates with different absorption, which could explain the differences in the amount of antigen used. Alternately, this could also be due to the expression system in which the recombinant protein was produced, specific posttranslational modifications, or differences in expressed protein lengths (N-terminal or C-terminal modified).

The performance of our iEIA was compared with the commercial assay PrioCHECK™ Porcine HEV Ab ELISA Kit (ThermoFisher Scientific, USA), which is based on plates coated with the recombinant HEV antigens 1 and 3. The results obtained with our iEIA are in high agreement with the results from commercial assays. We found 97.8% agreement with a *kappa* index 0.9399. We established a cut-off value of A492nm = 0.92 for the in-house ELISA, based on the ROC curve analysis (Figure 5). It is worth noting that the plant-produced HEV-3 ORF2 is suitable for use as a coating antigen in detecting HEV serum antibodies in humans [15] and pigs, with better sensitivity and specify compared to the prokaryotic cell expression system [17]. This demonstrates one of the advantages of plant expression systems: their ability to perform eukaryotic post-translational modifications. This and other advantages, such as safety, scalability, and low-cost production, have made plants a viable alternative to other expression systems [62]. The limitations of plant expression systems, such as proteolytic degradation of recombinant proteins and product heterogenecity due to the glycosylation pattern, have been overcome using different methods [63,64,65,66].

In Bulgaria, the production of valuable recombinant proteins in plants is hardly used in practice. This project aims to offer an alternative to the commercially available systems for diagnostic antigen production, as well as to validate a diagnostic test based on the plant-derived HEV-3 ORF2 protein. The ultimate goal is to produce this diagnostic kit in Bulgaria at a low enough price to allow routine testing of pigs and evaluation of HEV seroprevalence.

## 5. Conclusions

This study confirmed that plant-derived HEV-3 ORF2 diagnostic antigen could provide a cost-effective and scalable alternative to commercially available coating proteins. The development of this assay aims to provide reliable, consistent, and affordable serological detection of HEV in pigs, which could provide more readily available testing, leading to more accurate assessment of HEV prevalence in Bulgaria.

## Figures and Tables

**Figure 1 vaccines-09-00991-f001:**
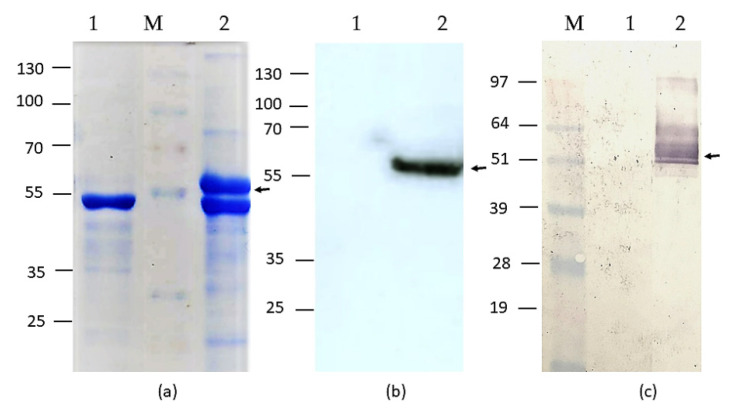
(**a**) SDS-PAGE of HEV-3 ORF2 110-610_6his protein. (**b**) Western blot with a monoclonal anti-HEV ORF2 ab. (**c**) Western blot with HEV-positive pig serum. M-pre-stained protein molecular marker (kDa), 1-total soluble protein extracted from plant leaves infiltrated with empty pEff. 2-total soluble protein extracted from plant leaves inoculated with pEff:HEV-3 ORF2 110-610_6his construct.

**Figure 2 vaccines-09-00991-f002:**
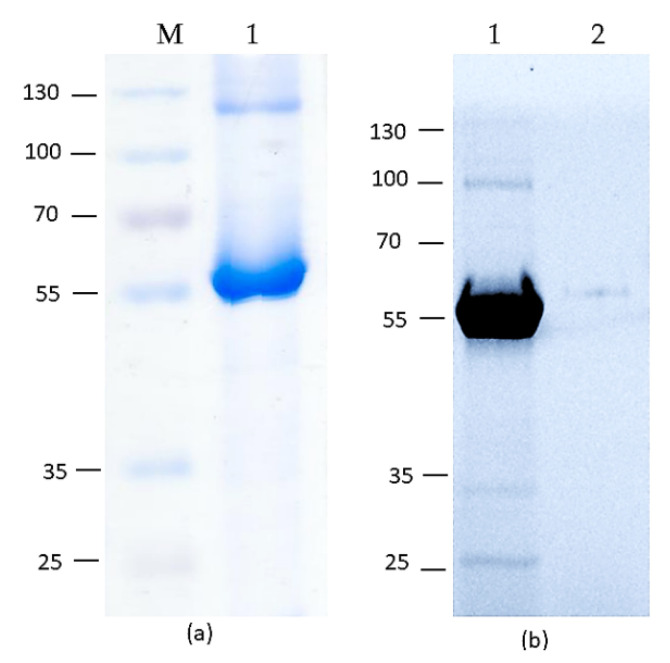
(**a**) SDS-PAGE of the purified HEV-3 ORF2 110-610_6his protein by immobilized metal-anion chromatography. (**b**) Western blot with an anti-HEV ORF2 mAb. M-presented protein molecular marker (kDa), 1-purified and dialyzed HEV-3 ORF2 110-610_6his protein, 2-protein extracted from non-inoculated plant leaves.

**Figure 3 vaccines-09-00991-f003:**
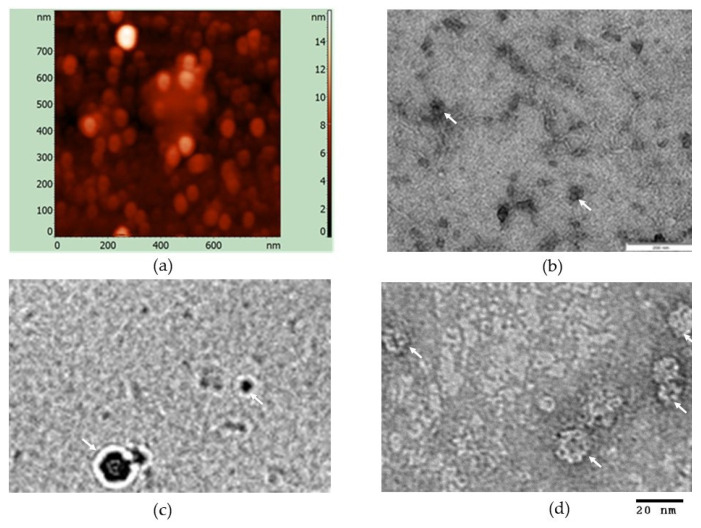
Analysis of the HEV-3 ORF2 110-610_6his protein. (**a**) Atomic force microscopy. (**b**) Transmission electron microscopy, scale bar = 200 nm. (**c**) Cryo-EM micrograph of hepatitis E virus particles. (**d**) TEM, scale bar = 20 nm; Arrows indicate VLPs.

**Figure 4 vaccines-09-00991-f004:**
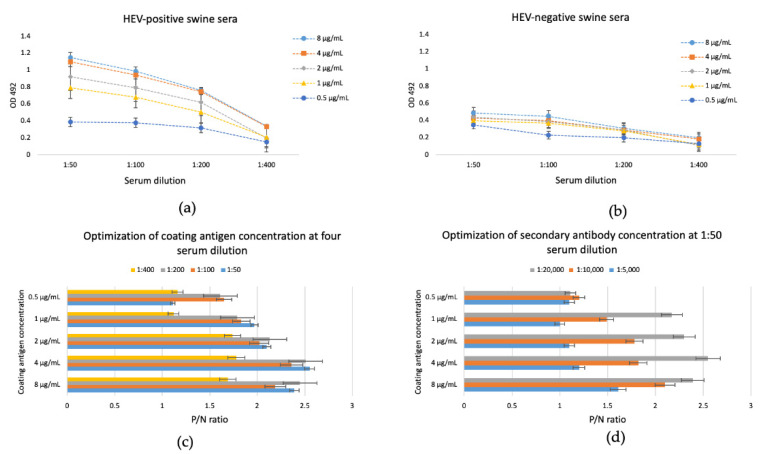
Determination of optimal iEIA conditions. Five positive and five negative serum samples previously determined with commercial kit were used. (**a**) Optimal dilutions of positive serum samples. (**b**) Optimal dilutions of negative serum samples; Data are shown as absorbance values of the used serum samples; the bars refer to standard error. The presented absorbance values are average values with standard errors calculated from five different negative serum and five different positive serum, repeated in four independent experiments. (**c**) Optimal antigen concentration was determined. (**d**) Secondary antibody dilution was determined using an antigen concentration of 4 µg/mL, and serum dilution 1:50. P/N ratio in (**c**,**d**) was calculated at the relation A(λ492nm)/A(λ492nm) positive/negative control.

**Figure 5 vaccines-09-00991-f005:**
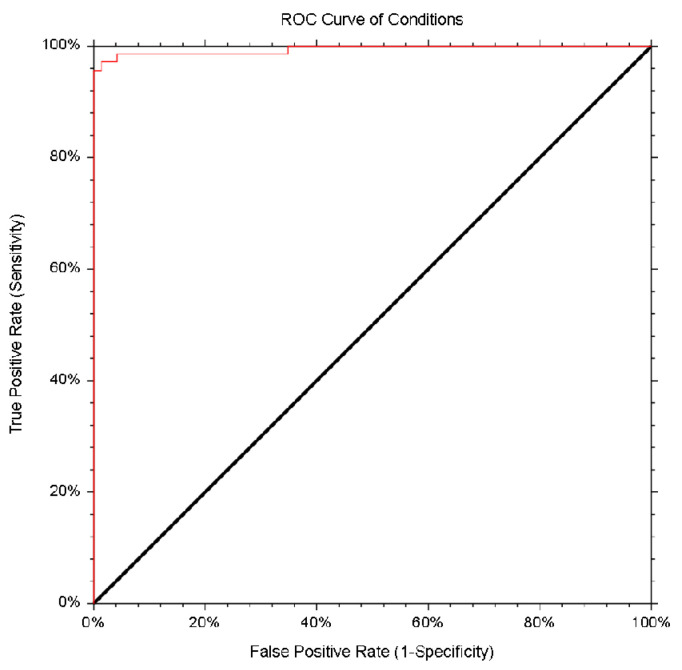
ROC analysis of the developed iEIA. ROC curve was generated using the results obtained by analyzing 72 negative and 69 positive pig serum samples, previously detected by PrioCHECK™ Porcine HEV Ab ELISA Kit. The red line shows the mean area under the curve (AUC) plot, with the AUC value of 0.9942 (95% Cl: 0.9672–0.999). NCSS 2021 Data Analysis software was used to perform the ROC analysis.

**Table 1 vaccines-09-00991-t001:** Performance agreement between the PrioCHECK™ Porcine HEV Ab ELISA commercial kit and the iEIA in the detection of anti-HEV IgG. 141 samples from pigs were selected and tested in duplicates with the commercial and iEIA tests.

		IN-HOUSE EIA
Commercial ELISA		**Positive**	**Negative**	**Total**
Positive	67	2	69
Negative	1	71	72
Total	68	73	141
Kappa indexSensitivitySpecificity	0.939997.1% (95% Cl: 89.9–99.65)98.6% (95% Cl: 92.5-99.96)

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
