# Peer review of "Development and Optimization of an Enzyme Immunoassay to Detect Serum Antibodies against the Hepatitis E Virus in Pigs, Using Plant-Derived ORF2 Recombinant Protein"

_vaccines, 2021, doi:10.3390/vaccines9090991_

Round 1

Reviewer 1 Report

Overall comments:

The authors describe an ELISA assay for the detection of antibodies against hepatitis E virus (anti-HEV) in pigs, using plant-derived recombinant HEV ORF2 capsid protein. The assay has been optimized and compared against a commercially available test (PrioCHECK® HEV Ab porcine, Prionics AG, Schlieren, Switzerland). The results show a 97.8% agreement between the two assays, indicating that the plant derived HEV antigen could be successfully used as a suitable, relatively inexpensive reagent. The authors use the receiver operating characteristic (ROC) curve, for describing and comparing the accuracy of the new in-house diagnostic test, including the optimal cut-off point. One potential issue with this approach is choosing the right, or most balanced cut-off which in this case is heavily dependent on the sensitivity and specificity of the test selected for comparison and may not be the most accurate reflection of the true value of the cut-off of the new in-house assay.

Specific comments:

  1. Materials and methods, lines 183-200; please describe how many positive/negative pig samples were used for the checkerboard optimization, how many of these were strong positive, average or weak for anti-HEV, what was the basis for their selection?
  2. Results, Fig.4c and lines 275-277; as presented the data shows that pig serum dilution of 1:200 or 1:100 is equivalent to the preferred 1:50. In fact there is no significant difference in terms of P/N ration for the three dilutions. This result may be very important as the background of the assay is relatively quite high and one possible way to reduce it is by increasing the starting dilution of the pig serum to an extent that this will not compromise the sensitivity. At the same time this will probably improve the specificity of the assay and decrease the cut-off values which is quite high as presented, almost 1.0 OD!
  3. The dilution of the secondary antibody (1:20,000) is reasonable starting at all concentrations of the antigen coating (from 1mg/ml to 8 mg/ml), however speaking of checkerboard titration there is no data for the different dilutions of the pig sera (only 1:50 shown).
  4. The ROC curve analysis including the selection of the cut-off is reasonable, however one should keep in mind the test the authors are comparing the new in-house assay; PrioCHECK®HEV Ab porcine, Prionics AG, Schlieren, Switzerland itself is not the gold standard in anti-HEV diagnostics. This test claims a sensitivity of 91% and specificity of 94.1%. Therefore, it is a good idea to improve the cut-off of the new assay by reducing the background for once rather than rely on the ROC selection of the cut-off based on the comparison of the two assays.

Author Response

Comment 1: Materials and methods, lines 183-200; please describe how many positive/negative pig samples were used for the checkerboard optimization, how many of these were strong positive, average or weak for anti-HEV, what was the basis for their selection?

Answer1: We began the optimization of our EIA unaware if our positive samples were strong, average or weak.  Initially, five positive and five negative samples were randomly selected from a large library of samples which did not contain the information if the positive samples were strong, average or weak.  Consequently, based on follow up testing we determined that we have 1 strong, 3 average, and 1 weak positive samples.

We included the following text on line 187:

[187] A total of 5 negative and 5 positive serum samples, randomly selected from previously evaluated samples, were tested in three independent experiments.  

Comment 2: Results, Fig.4c and lines 275-277; as presented the data shows that pig serum dilution of 1:200 or 1:100 is equivalent to the preferred 1:50. In fact there is no significant difference in terms of P/N ration for the three dilutions. This result may be very important as the background of the assay is relatively quite high and one possible way to reduce it is by increasing the starting dilution of the pig serum to an extent that this will not compromise the sensitivity. At the same time this will probably improve the specificity of the assay and decrease the cut-off values which is quite high as presented, almost 1.0 OD!

Answer 2: We agree with Reviewer 1 that there is no significant difference between the results from the 1:50 and 1:200 serum dilutions. In both cases, the P / N ratio is approximately 2.5.  We did consider using the 1:200 dilution, but in our repeated optimization reactions we noticed that weakly positive samples, at 1:200 serum dilutions resulted in a significantly lower optical density, which increased the chance of reporting such samples as negative.  The results from the 1:200 dilution also gave much highte error, most likely because of the lower sensitivity for weakly positive samples.  To avoid the exclusion of samples with weak positivity, we decided to use the serum dilution of 1:50.  We knew that it will give a higher background, but to counteract that effect, we put extra efforts in optimizing the blocking agent and the concentration of the secondary antibody. Bringing down the background of the tested samples, as much as possible, was one of our biggest objectives. We first optimized the coating buffer (Bicarbonate/carbonate coating buffer (100 mM) pH 9.6 compared with PBS pH 7.6) and then the blocking buffer (comparing PBST–3% [w/v] BSA with PBS- 5%[w/v] NFSM, 1%[w/v] BSA). We found that the Bicarbonate/carbonate coating buffer (100 mM) pH 9.6 gives better results compare to PBS pH 7.6. For the blocking buffer, we obtained optimal results when we used a PBST–3% [w/v] BSA buffer. We found that the use of NFDM as a blocking agent gives a very high background and non-specific binding.  We chose to work with the 1:50 dilution because it gave the best results based on our optimization conditions.  We decided not to use the 1: 200 dilution, in order to avoid false negative results.

Comment 3: The dilution of the secondary antibody (1:20,000) is reasonable starting at all concentrations of the antigen coating (from 1mg/ml to 8 mg/ml), however speaking of checkerboard titration there is no data for the different dilutions of the pig sera (only 1:50 shown).

Answer 3: The secondary antibody titration results are not presented in Figure 4 because we did not want to show a figure with too many element which could make it confusing and more difficult to understand. Once we determined and chose to use 4µg/ml as the coating antigen concentration and 1:50 as the serum dilution, we concluded that it would be most relevant to show the obtained results from the two above mentioned parameters when used with different concentrations of the secondary antibody.  For this reason and to keep the figure more focused and clear, we decided not to include all of the secondary antibody titration results.  However, if requested, we will be happy to provide these result.

Comment 4: The ROC curve analysis including the selection of the cut-off is reasonable, however one should keep in mind the test the authors are comparing the new in-house assay; PrioCHECK®HEV Ab porcine, Prionics AG, Schlieren, Switzerland itself is not the gold standard in anti-HEV diagnostics. This test claims a sensitivity of 91% and specificity of 94.1%. Therefore, it is a good idea to improve the cut-off of the new assay by reducing the background for once rather than rely on the ROC selection of the cut-off based on the comparison of the two assays.

Answer 4. We thank Reviewer 1 for the abovermentioned recommendaions.  The background of our results is indeed higher than what is accepted as optimal, and it is necessary to further optimize the conditions if and when the in-house ELISA were to become a routinely used method for anti-HEV antibody detection. Reducing the cut-off point and background are main tasks in our future plans.  However, in defense of our method, we have shown that our assay does not generate a significantly high number of false positive samples, which one might expect from reactions with a high background.  Only one of the negative samples determined by the commercial kit, was labeled as positive, which suggests, at least based on our chosen validation method, that the conditions we have chosen are working appropriately.  

We also agree that the commercial kit we used in our validation study is not the gold standard in anti-HEV diagnostics.  We chose it because in practical terms, on the field, in Bulgaria, that is the kit primarily used for detecting anti-HEV antibodies.  Showing that our assay performs just as well as the commercially available kit, or perhaps we could improve it to work even better, carries great importance for our future objectives.

Reviewer 2 Report

The document presents results that can be scalable for obtaining a test with sufficient validity to be used in the field against HEV. 

line 111: It is suggested to write the accession number of the sequence entered in the transformation cassette.

line 327-328: References 

Author Response

Dear Reviewer,

Thank you very much for your comments.

line 111: It is suggested to write the accession number of the sequence entered in the transformation cassette.

Now  [ 114] The target HEV-3 ORF2 110–610 gene (GenBank accession number DQ079627.1)  was produced as previously described by our group

line 327-328: References

We include 3 references.

[328] Ponterio, E., Di Bartolo, I., Orrù, G. et al. Detection of serum antibodies to hepatitis E virus in domestic pigs in Italy using a recombinant swine HEV capsid protein. BMC Vet Res 10, 133 (2014). https://doi.org/10.1186/1746-6148-10-133; Lee WJ, Shin MK, Cha SB, Yoo HS. Development of a novel enzyme-linked immunosorbent assay to detect anti-IgG against swine hepatitis E virus. J Vet Sci. 2013;14(4):467-72. doi: 10.4142/jvs.2013.14.4.467. Epub 2013 Dec 19. PMID: 24421718; PMCID: PMC3885741.

Reviewer 3 Report

General

This article is about the design and performance evaluation of an immunoassay for detecting hepatitis E antibodies in the serum of pigs, using plant-produced recombinant HEV-3 ORF2 as an antigenic coating protein. The authors have a background in the field. Interestingly they exploit an innovative concept based on the use of a plant transient expression.This concept has led to several publications in collaboration with George  P. Lemonossoff who is a named inventor on granted patent WO 29087391 A1 (see Ann Lab. Med. 2017). This is for the past and background. As a consequence this article is merely confirmatory of previous studies. This immunoassay is a competitor to other commercially available immunoassays. Unfortunately the advantages of this immunoassay compared with other immunoassays are not clear. If the goal is to provide a commercially available immunoassay, the authors have to deal with existing patents. As to wether this article provides new information with respect to  a previous publication (Mazalovska et al. Ann. Lab Med 37, 2017) has remained unknown.

Specific

As it stands this article poses a potential concern in terms of conflict of interest. For example the sentence (line 103) "Recently, we transiently expressed and characterized the recombinant HEV-3...." is incorrect as the list of authors of the 2 papers is not the same. The sentence should be reworded in the passive form, and the "we" omitted.  In the same way, see line 342 "In our previous study, we described...."

Author Response

Dear reviewer, thank you for the critical evaluation. This work is a continuation of our work with George Lomonossoff. George Lomonossoff owns a patent right on the pEAQ-HT vector (patent 9087391 A1), but not on transient expression of HEV ORF2 in Nicotiana benthamiana plants. I (G. Zahmanova) can declare that we do not have a conflict of interest with George Lomonossoff.  In Bulgaria, we are currently working on launching a procedure for registration of a utility model for detection of antibodies against the Hepatitis E virus in pigs, using plant-derived ORF2 recombinant protein.  I agree, this work confirms our previous observagtions from 2017, but it also adds to it. Both systems (pEAQ- HT and pEff) have been used for the expression of the HEV ORF2 capsid protein, and the pEff vector showed a higher level of accumulation of the HEV capsid protein.  In this study, we used the pEff vector for the expression of ORF2 and the IMAC method applied here obtained ~200 µg HEV ORF2 / g of fresh tissue, which is two times more than what we observed in the 2017  study ( Ann. Lab. Med. 2017). Also, here we demonstrate that the HEV capsid protein formed VLPs.  The use of HEV particulate capsid protein as a coating antigen gives us advantages, such as stability and detection of antibodies interacting with the conformational epitopes. The plant-derived HEV ORF2 capsid protein   remains stable and retains its antigenic activity for more than a year. We have not commented on this fact in the article, but we have established it in our work. We validated the in-house ELISA, comparing its sensitivity and specificity with a commercial kit. We use the receiver operating characteristic (ROC) curve, for describing and comparing the accuracy of the new in-house diagnostic test.  We  also optimized the conditions for conducting the in-house ELISA and  determined the optimal cut-off point. The above gave us reason to offer this work for publication.

We agree with the specific remarks and they will be changed.

[104] Recently, the recombinant HEV-3 ORF2 110-610_6his was transiently expressed and characterized in plants, showing its immunogenic [48] and diagnostic potential [15]

[347] Our previous studies describe the successful expression of the HEV-3 ORF2 110-610 gene [37,48] using the Cowpea Mosaic Virus (CPMV)-based vector pEAQ-HT [Sainsbury 2008 and 2009] and the potato X virus (PVX)-based vector pEff [41].

Round 2

Reviewer 3 Report

In the response to the reviewer the senior corresponding author clearly declares no conflict of interest with G. Lomonossoff. Hence a major barrier is removed. The authors should be more precise in their discussion about the stability over time of the antigenic activity of the plant-derived HEV ORF2 capsid protein.

Author Response

We thank Reviewer 3 for all comments and suggestions. In response to his comments, we added the following sentences to the discussion:

[379] We also observed that purified HEV ORF2 can retain its antigenicity for more than a year when stored at 4 ° C. The antigenicity of a newly purified and purified more than a year ago HEV ORF2 proteins was compared by ELISA (data not shown). Indeed, the one-year-old protein did show decreased antigenicity, but the observed difference was not significant, suggesting that the antigenic potency and functionality of the purified plant HEV ORF2 remains high even after a year-long storage at 4 ° C.